# The Endemic Species Flock of *Labeobarbus* Spp. in L. Tana (Ethiopia) Threatened by Extinction: Implications for Conservation Management

Shewit Gebremedhin [1,2,*], Stijn Bruneel [1] , Abebe Getahun [3], Wassie Anteneh [4,5] and Peter Goethals [1]

1   Department of Animal Science and Aquatic Ecology, University of Ghent, 9000 Ghent, Belgium; Stijn.Bruneel@ugent.be (S.B.); Peter.Goethals@UGent.be (P.G.)
2   Department of Fisheries, Wetlands and Wildlife Management, Bahir Dar University, Bahir Dar 6000, Ethiopia
3   Department of Zoological Sciences, Addis Ababa University, Addis Ababa 1000, Ethiopia; zoology_cns@aau.edu.et
4   Department of Biology, Bahir Dar University, Bahir Dar 6000, Ethiopia; Wassie.Melkamu@fao.org
5   Food and Agriculture organization, Addis Ababa 1000, Ethiopia
*   Correspondence: shewitgebremedhin.kidane@ugent.be; Tel.: +251-92-051-8412

**Abstract:** The endemic *Labeobarbus* species in Lake Tana are severely affected by anthropogenic pressures. The implementation of fisheries management is, therefore, vital for their sustainable exploitation. This study aimed at investigating the catch distribution and size at 50% maturity ($FL_{50\%}$) of the *Labeobarbus* species. Samples were collected monthly from May 2016 to April 2017 at four sites. The relative abundance, catch per unit effort (CPUE), and size distribution of these species was computed, and logistic regression was used to calculate $FL_{50\%}$. Of the 15 species observed, five species constituted 88% of the total catch. The monthly catch of the *Labeobarbus* spp. declined by more than 85% since 1993 and by 76% since 2001. Moreover, the CPUE of *Labeobarbus* has markedly decreased from 63 kg/trip in 1991–1993 to 2 kg/trip in 2016–2017. Additionally, large size specimens (≥30 cm fork length) were rarely recorded, and $FL_{50\%}$ of the dominant species decreased. This suggests that the unique species flock may be threatened by extinction. Given the size distribution of the species, the current social context, and the need for a continuous supply of fish for low-income communities, a mesh-size limitation represents a more sustainable and acceptable management measure than a closed season. This paper illustrates the tension between sustainable development goal (SDGs) 1—No Poverty, 2—Zero Hunger, and 8—Decent Work and Economic Growth in Bahir Dar City on the one hand, and SDG's 11—Sustainable Cities and Communities, 12—Responsible Consumption and Production, and 14—Life Below Water on the other hand. A key for the local, sustainable development of the fisheries is to find a balance between the fishing activities and the carrying capacity of the Lake Tana. Overfishing and illegal fishing are some of the major threats in this respect.

**Keywords:** fisheries management; fish stock; illegal fishing; size at maturity

## 1. Introduction

Lake Tana, Ethiopia's largest lake, harbors 27 fish species, which belong to four families: Cichlidae, Clariidae, Nemacheilidae, and Cyprinidae. The first three families are represented by a single species each, whereas, Cyprinidae comprises 24 species, including *Labeobarbus* species [1]. With 17 endemic *Labeobarbus* spp. [1], the lake harbors the highest number of endemic cyprinid species in Ethiopia. *Labeobarbus* spp. are the most abundant in terms of biomass, and nine of them are known to migrate for spawning [2–5].

Due to their functional diversity, *Labeobarbus* spp. play a vital role in regulating the structure and functioning of the lake ecosystem. Due to their wide distribution in the lake and varying feeding habits, *Labeobarbus* spp. play an important role in food-web interactions. Eight of the 17 species are piscivorous, while the remaining feed on phytoplankton, zooplankton, and detritus [6]. Additionally, *Labeobarbus* spp. provide the basis for food, employment, and income for thousands of people. Due to their relatively large size (max. length > 40 cm), the majority of the *Labeobarbus* spp. are highly valued fishery resources and constitute the base for annual fishers' income [7]. These species and other commercially important species are currently supporting the livelihoods of more than 90,000 people in the Lake Tana area [8].

Despite their ecological and socio-economic importance, *Labeobarbus* spp. generally, have been received little consideration in development decisions. Local managers, who are responsible for natural resources conservation, are often not aware of their importance. Consequently, detrimental pressures, contributing to the survival of these species, such as illegal fishing and environmental degradations, are often neglected. Strong regulations and their proper enforcement are paramount to protect the endemic *Labeobarbus* spp. and their attributes, but these are virtually non-existent in Lake Tana. Ethiopia has endorsed fish resource development and the utilization of proclamations in 2003 and fisheries resource development protection and utilization proclamation enforcement in 2007. However, these proclamations are not yet properly implemented. Currently, 98% of the fishers in Lake Tana use illegal fishing gears, such as the legally banned monofilaments [9], and they deliberately undertake fishing activities during the spawning seasons and target the spawning grounds of the migratory *Labeobarbus* spp. [3]. Since direct pressures have not received attention, it is no surprise that indirect pressures remain unaddressed. Environmental degradation of spawning grounds and nursery habitats due to damming, sand mining, and recession agriculture have deleterious effects on the endemic *Labeobarbus* spp. [10,11]. At the moment, dams are being constructed without fish-ways. In this state, the dams are seriously disrupting the migrating fish populations and reducing fish biodiversity.

The rapidly intensifying pressures on the Lake Tana ecosystem have resulted in a decrease in the abundance and diversity of the *Labeobarbus* spp. [11–13]. The catch per unit effort (CPUE) of the *Labeobarbus* spp. has decreased drastically from 63 kg/trip in 1991–1993 [12] to less than 6 kg/trip in 2010 [14]. Additionally, six of the endemic *Labeobarbus* spp. have already been reported in the International Union for Conservation of Nature (IUCN) red list of threatened species [15,16]. The present study aims to support the conservation management of these species. We had two main objectives: (1) to describe and quantify the abundance, catch, size distribution, and size at first maturity of *Labeobarbus* spp., and (2) to recommend optimal management measures based on these results, needed to combat the conservation challenges of these species.

## 2. Materials and Methods

### 2.1. Study Area

Lake Tana originated two million years ago by volcanic blocking of the Blue Nile River [17]. The lake contains half of the country's surface freshwater and has seven perennial and approximately 60 intermittent tributaries. Its watershed is situated at the basaltic plateau of the north-western highlands of Ethiopia at 12° N, 37°15′ E at about 1800 m altitude above sea level. Lake Tana, the headwater of the Blue Nile River, is a shallow lake with an average depth of 8 m and a maximum depth of 14 m. The lake has an area of about 3050 km$^2$. High waterfalls (40 m) isolate the lake from the lower reaches of the Blue Nile River at Tississat (smoking water). Lake Tana has temperatures ranging from 16.4 °C to 31.2 °C and is one of the most important development corridors for the national economy since it has the potential for irrigation, hydroelectric power, water supply, high-value crops, livestock, fish production, and eco-tourism.

## 2.2. Sampling Techniques

Four sites, the outflow of the Blue Nile River (Bahir Dar site = 1), the river mouths of Gumara (Gumara site = 2), Dirma (Gorgora site = 3), and Gilgel Abay (Kunzla site = 4) (Figure 1) were sampled. Fish were collected each month from May 2016 to April 2017. Ten multifilament gillnets (50 m length by 1.5 m depth), two each with 6, 8, 10, 12, and 14 cm stretched mesh size, were used. Multifilament-gillnets were grouped into two. Five-gillnets, one from each mesh size, were connected end to end to form a 250 m long and a 1.5 m wide panel. To increase the probability of capturing fish from different habitats, one group of combined gillnets was set an estimated 1 km from the shore area and the other group at an estimated 0.5 km farther offshore from the first group of gillnets. Gillnets were fished for 16 hours, from around 4:00 pm until 8:00 am and immersed to the bottom using anchors. Monofilament gillnets (100 × 1.5 m of 4 and 6 cm stretched mesh size) were used to sample small sized fishes. The monofilament gillnets were set in the shore areas of the lake during daytime from 8:00–10:00 am. The effort was standardized as the number of nets set per trip (nets set/trip). Captured *Labeobarbus* individuals were identified to species using keys [1] and subsequently grouped by mesh size. Specimens with complex features were difficult to identify and considered hybrids. These fish were merged with the *Labeobarbus intermedius* group. Fork length of each specimen (to the nearest 0.1 cm) and total weight (to the nearest 0.1 g) were measured in the field using a measuring board and a balance. After dissection, each fish was sexed, and the gonad maturity determined using a seven-point maturity scale [18]. The majority of the *Labeobarbus* spp. caught by gillnets died, but those caught without major injury were quickly identified and returned to the lake.

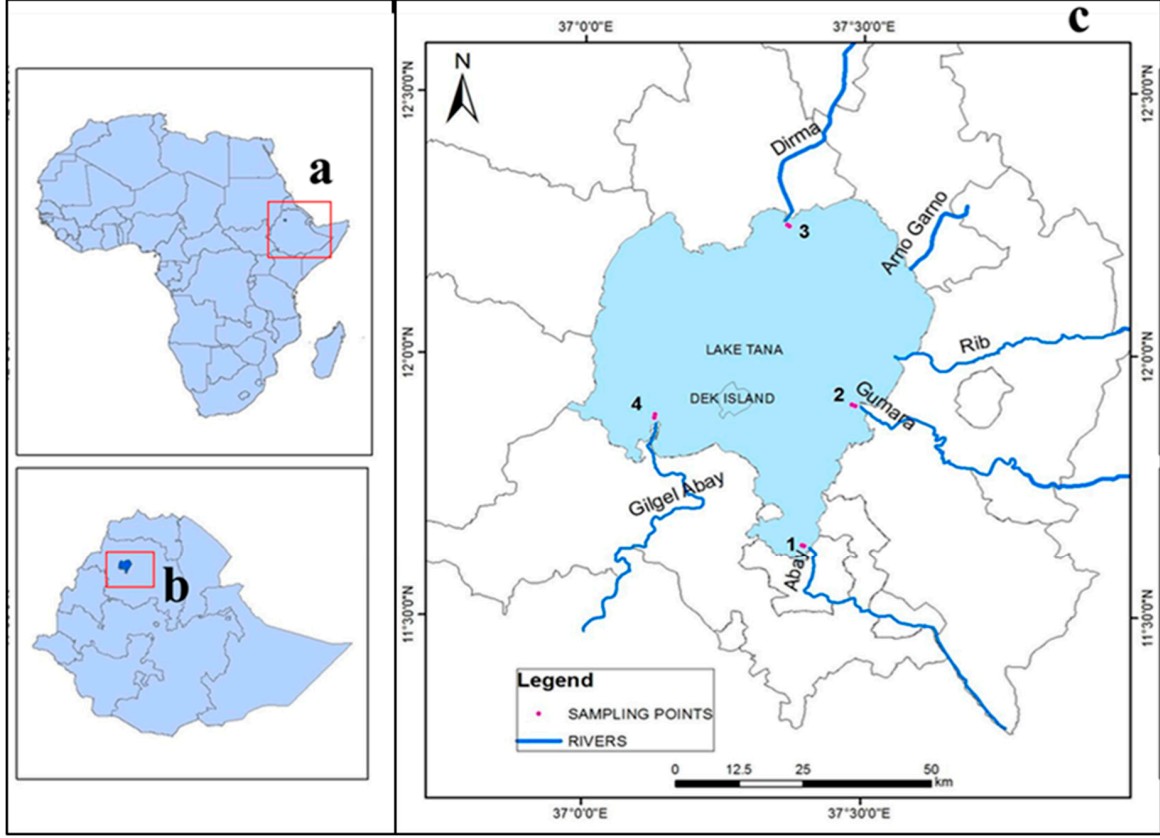

**Figure 1.** (**a**,**b**) red rectangles indicate the location of Ethiopia and Lake Tana in the Maps of Africa and Ethiopia, respectively. (**c**): Map of the Lake Tana watershed showing the study sites; pink dots are sampling sites and the numbers are the name of sites (see in the text).

*2.3. Data Analysis*

Descriptive statistics were used to determine the mean weight and length frequency of the catch. The distribution of *Labeobarbus* spp. across sampling sites were tested using the Chi-square test of independence (10 species and four sites). Potential differences in the distribution among study sites were tested using Z-test based on the Holm–Bonferroni method (adjusted *P*-value). The index of relative importance was computed as IRI = (%N + %W) × %F [19], where %N and %W are the percentages in number and weight of each species in the total catch, and %F is the percentage of occurrence per gillnet (% of all net nights containing a given species).

Data were not collected from the Gumara site in August 2016. Hence, the average catch in July and September at Gumara was used for the analysis instead. Catch per unit effort (CPUE in kg wet weight/trip), size at maturity, and length distribution were determined only for the four dominant *Labeobarbus* spp. CPUE was calculated as $CPUE = \frac{\sum C_i}{\sum f_i}$, where $C_i$ is the ith catch expressed in weight and $f_i$ is its respective fishing effort [20]. The effect of sampling sites, seasons, and their interaction on the catch of the dominant *Labeobarbus* was tested using Aligned ranks transformed ANOVA and R-software (version 3.5.1, R Developer Core Team, R Foundation for Statistical Computing, Vienna, Austria). The pairwise comparison was made using Tukey's method. The factor season was categorized into two levels (wet and dry seasons). The months June–November were classified as the wet season, while December–May was classified as the dry season. The classification of seasons was based on the spawning period of the species and the rainfall availability.

Length of fish collected during the breeding season (July to October) was used to determine $FL_{50\%}$ the length at which 50% of the males and females attain maturity. Maturity (recorded as "0" for immature and "1" for mature) was the dependent variable, while the individual length (continuous) was the independent variable. Gonads from stage-IV (with large white testes for males and large and full ovary for females) onwards were considered mature when determining the fraction of mature fish at different lengths. Female and male fishes were analyzed separately. The proportion of mature fish (P) was estimated as:

$$P = \frac{\text{Number of mature fish}}{\text{Total number of fish}}$$

A logistic function was used to describe the fraction of mature fish versus length interval. The logistic regression was fitted in R software version 3.5.1 with the general linear model procedure. The logistic equation is described [21] as $P = \frac{1}{(1+e(a+bFL))}$. Where "P" is proportion mature fish in length class L, "a" is an intercept, "b" is the slope, and "FL" is fork length. The confidence intervals for the parameters of the logistic regression were estimated via bootstrapping with the boot case function, and the number of bootstrap samples was a thousand. The probability of being mature at a given value of the explanatory variable (fork length) was computed as $P = \frac{e^{a+bFL}}{1+e^{a+b}}$. A figure showing the fitted logistic regression line was constructed with the fit-Plot function from the FSA package [22]. The length at which 50% of the fish population has reached maturity was computed as $FL_{50\%} = \frac{-a}{b}$. Fish lengths were grouped at intervals of 5 cm, and the number of individuals for each length group computed. Mid-length of each length group was calculated and plotted against the number of individuals in each length group.

## 3. Results

*3.1. Abundance and Catch Distribution of Labeobarbus Species*

3.1.1. Relative Abundance

During this study, 4235 specimens consisting of 15 *Labeobarbus* spp. were recorded from the study sites (Table 1). Five species (*L. intermedius*, *L. tsanensis*, *L. platydorsus*, *L. megastoma*, and *L. brevicephalus*), constituted 88% of the total catch in number, whereas, the remaining 10 species constituted 12%. No *Labeobarbus osseensis* was caught during the sampling period, and less than 14 specimens were collected

of *L. dainellii*, *L. gorguari*, and *L. acutirostris*. *Labeobarbus dainellii* was absent at Kunzla and Gumara sites and no *L. gorguari* and *L. acutirostris* were recorded at Gumara and the Bahir Dar sites, respectively. The relationship between the relative abundance of the *Labeobarbus* spp. and study sites was significantly different (Chi-square = 385, df = 42, P = 0.000, Appendix A). The overall catch (kg) of *Labeobarbus* spp. was highest from July to September when migratory *Labeobarbus* spp. form spawning aggregations (Figure 2).

**Table 1.** The relative abundance of *Labeobarbus* spp. from exploratory gillnet catches by number (N), weight (W), the frequency of occurrence per gillnet (%F), and the index of relative importance. IRI is the index of relative importance, %N and %W are the percentages in number and weight of each species in the total catch, and %F is the percentage of occurrence per gillnet.

| Species | N | %N | W (kg) | %W | %F | IRI | %IRI |
|---|---|---|---|---|---|---|---|
| *L. intermedius* | 1454 | 34 | 241 | 27 | 100 | 6100 | 35 |
| *L. tsanensis* | 837 | 20 | 162 | 18 | 100 | 3800 | 22 |
| *L. platydorsus* | 623 | 15 | 171 | 19 | 100 | 3400 | 19 |
| *L. megastoma* | 339 | 8 | 131 | 15 | 71 | 1643 | 9 |
| *L. brevicephalus* | 446 | 11 | 31 | 3 | 57 | 800 | 5 |
| *L. crassibarbis* | 147 | 3 | 51 | 6 | 57 | 514 | 3 |
| *L. nedgia* | 98 | 2 | 28 | 3 | 86 | 429 | 2 |
| *L. gorgorensis* | 78 | 2 | 17 | 2 | 65 | 262 | 1 |
| *L. surkis* | 72 | 2 | 7 | 1 | 48 | 143 | 1 |
| *L. truttiformis* | 68 | 2 | 34 | 4 | 43 | 257 | 1 |
| *L. longissimus* | 28 | 1 | 11 | 1 | 38 | 76 | 0 |
| *L. macrophtalmus* | 26 | 1 | 8 | 1 | 43 | 86 | 0 |
| *L. acutirostris* | 13 | 0 | 6 | 1 | 24 | 31 | 0 |
| *L. gorguari* | 4 | 0 | 1 | 0 | 14 | 1 | 0 |
| *L. dainellii* | 2 | 0 | 1 | 0 | 5 | 0 | 0 |
| Total | 4235 | | 899 | | | 17,542 | 100 |

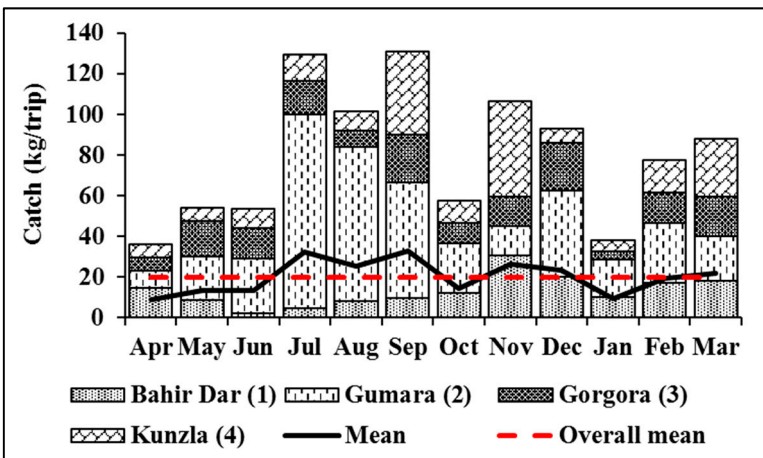

**Figure 2.** Distribution of *Labeobarbus* spp. catch (kg)/trip (overall catch, y-axis) as a function of time (month, x-axis) from experimental fisheries during the period of May 2016 to April 2017. The solid black line indicates the mean of the catch each month, while the dotted red line is the overall mean of the yearly catch.

*Labeobarbus* intermedius was frequently caught in gillnets and contributed the most to the overall number and weight of the total catch (Table 1). The index of relative importance was highest for *L. intermedius* followed by *L. tsanensis*, *L. platydorsus*, and *L. megastoma*, respectively (Table 1), and these four species comprised 85% of the IRI value. Five *Labeobarbus* spp. (*L. longissimus*, *L. macrophtalmus*,

*L. acutirostris*, *L. gorguari*, and *L. dainellii*) contributed less than one percent to the IRI value each (Table 1).

### 3.1.2. Catch Per Unit Effort

The catch per unit effort (CPUE in kg-wet weight) of four dominant *Labeobarbus* spp. at the four-study sites and during two seasons (wet and dry) was computed (Figure 3). The catch difference among sampling sites was significantly different for *L. megastoma* (F = 9.30, df = 3, P = 0.0001), while this was not different for *L. intermedius*, (F = 2.67, df = 3, P = 0.061), *L. tsanensis* (F = 2.67, df = 3, P = 0.06), and *L. platydorsus* (F = 2.28, df = 3, P = 0.093) (Table 2). For *L. intermedius*, the catch difference between the seasons was significant (F = 10.04, df = 1, P = 0.003), but there was no significant difference for *L. tsanensis* (F = 1.17, df = 1, P = 0.285), *L. platydorsus* (F = 0.014, df = 1, P = 0.906), and *L. megastoma* (F = 10.886, df = 1, P = 0.352) (Table 2). The interaction among sampling sites and seasons was significant for *L. intermedius* (F = 4.37, df = 3, P = 0.009), while this was not significant for *L. tsanensis* (F = 2.07, df = 3, P = 0.120), *L. platydorsus* (F = 1.04, df = 3, P = 0.387), and *L. megastoma* (F = 2.80, df = 3, P = 0.052). The interaction between sites and seasons was significant only for *L. intermedius* and the pairwise analysis revealed that the catch difference between BD-GU and GO-GU during the dry season were significantly different from the catch difference between BD-GU and GO-GU during the wet season, respectively (Table 3).

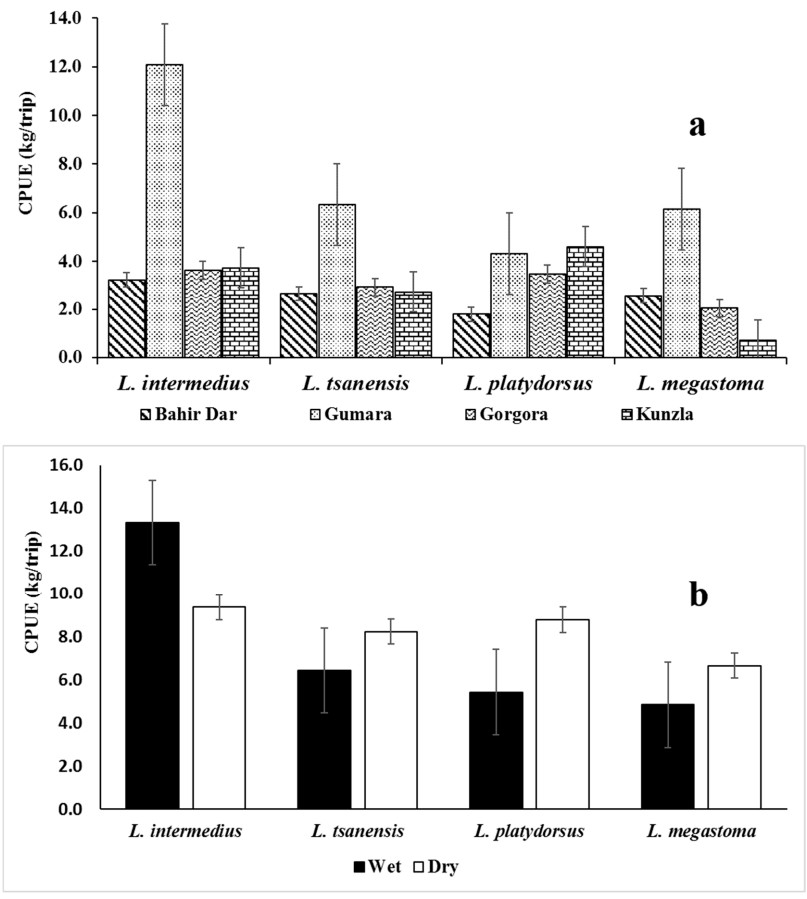

**Figure 3.** The distribution of CPUE of the four dominant *Labeobarbus* spp. across the four study sites (**a**) and wet and dry seasons (**b**) in Lake Tana during 2016–2017.

**Table 2.** The catch difference among sampling sites and seasons for four *Labeobarbus* species from Lake Tana. The data were collected from May 2016–April 2017 and analyzed using the Aligned rank transformed ANOVO. The degree of freedom was 40. The BD is Bahir Dar, GO is Gorgora, GU is Gumara, and KU is Kunzla.

| | *L. intermedius* | | | | *L. tsanensis* | | | |
|---|---|---|---|---|---|---|---|---|
| Contrast | estimate | SE | t-ratio | *P*-value | estimate | SE | t-ratio | *P*-value |
| BD-GO | −0.29 | 5.29 | −0.06 | 0.999 | −3.75 | 5.57 | −0.67 | 0.907 |
| BD-GU | −12.83 | 5.29 | −2.43 | 0.089 | −13.75 | 5.57 | −2.47 | 0.081 |
| BD-KU | −1.88 | 5.29 | −0.35 | 0.985 | −1.17 | 5.57 | −0.21 | 0.997 |
| GO-GU | −12.54 | 5.29 | −2.37 | 0.099 | −10.00 | 5.57 | −1.80 | 0.291 |
| GO-KU | −1.58 | 5.29 | −0.30 | 0.991 | 2.58 | 5.57 | 0.46 | 0.967 |
| GU-KU | 10.96 | 5.29 | 2.07 | 0.180 | 12.58 | 5.57 | 2.26 | 0.125 |
| Dry-Wet | −11.80 | 3.71 | −3.17 | 0.003 | −4.67 | 4.31 | −1.08 | 0.29 |
| | *L. platydorsus* | | | | *L. megastoma* | | | |
| BD-GO | −11.92 | 5.62 | −2.12 | 0.165 | −4.50 | 4.69 | −0.96 | 0.773 |
| BD-GU | −13.17 | 5.62 | −2.34 | 0.106 | −17.25 | 4.69 | −3.68 | 0.004 |
| BD-KU | −10.25 | 5.62 | −1.82 | 0.278 | 6.75 | 4.69 | 1.44 | 0.483 |
| GO-GU | −1.25 | 5.62 | −0.22 | 0.996 | −12.75 | 4.69 | −2.72 | 0.046 |
| GO-KU | 1.67 | 5.62 | 0.30 | 0.991 | 11.25 | 4.69 | 2.40 | 0.094 |
| GU-KU | 2.92 | 5.62 | 0.52 | 0.954 | 24.00 | 4.69 | 5.12 | 0.0001 |
| Dry-Wet | 0.50 | 4.21 | 0.12 | 0.906 | 3.92 | 4.16 | 0.94 | 0.35 |

**Table 3.** Pairwise comparison of the interaction effect of sites and seasons on catch of *Labeobarbus intermedius*. Data was collected from May 2016–April 2017 and analyzed using Aligned rank transformed ANOVO. Degree of freedom was 40. BD is Bahir Dar, GO is Gorgora, GU is Gumara, and KU is Kunzla.

| Site-Pairwise | Season-Pairwise | Estimate | R-Ratio | *P*-Value |
|---|---|---|---|---|
| BD-GO | Dry-Wet | −2.33 | −0.24 | 0.811 |
| BD-GU | Dry-Wet | 29.00 | 2.99 | 0.005 |
| BD-KU | Dry-Wet | 12.00 | 1.24 | 0.223 |
| GO-GU | Dry-Wet | 31.33 | 3.23 | 0.003 |
| GO-KU | Dry-Wet | 14.33 | 1.48 | 0.147 |
| GU-KU | Dry-Wet | −17.00 | −1.75 | 0.087 |

*3.2. Size Structure of Labeobarbus Species*

3.2.1. Length Distribution

Majority of the specimens of *L. intermedius*, *L. tsanensis*, and *L. platydorsus* were in the length groups of 15–20 cm (18 mid-length) and 20–25 cm (23 mid-length), while specimens with a fork length of more than 33 cm were rarely caught (Figure 4). For *L. megastoma*, a high length-frequency was recorded for specimens included within the length groups of 25–30 cm (28 mid-length) and 30–35 cm (38 mid-length), whereas specimens larger than 38 cm were rare (Figure 4). Relatively high frequencies of small length groups were recorded during the breeding time of the *Labeobarbus* spp., while large specimens were rare in all seasons (Figure 5). We rarely observed specimens ≥ 35 cm fork length, which was also true for the commercial catch.

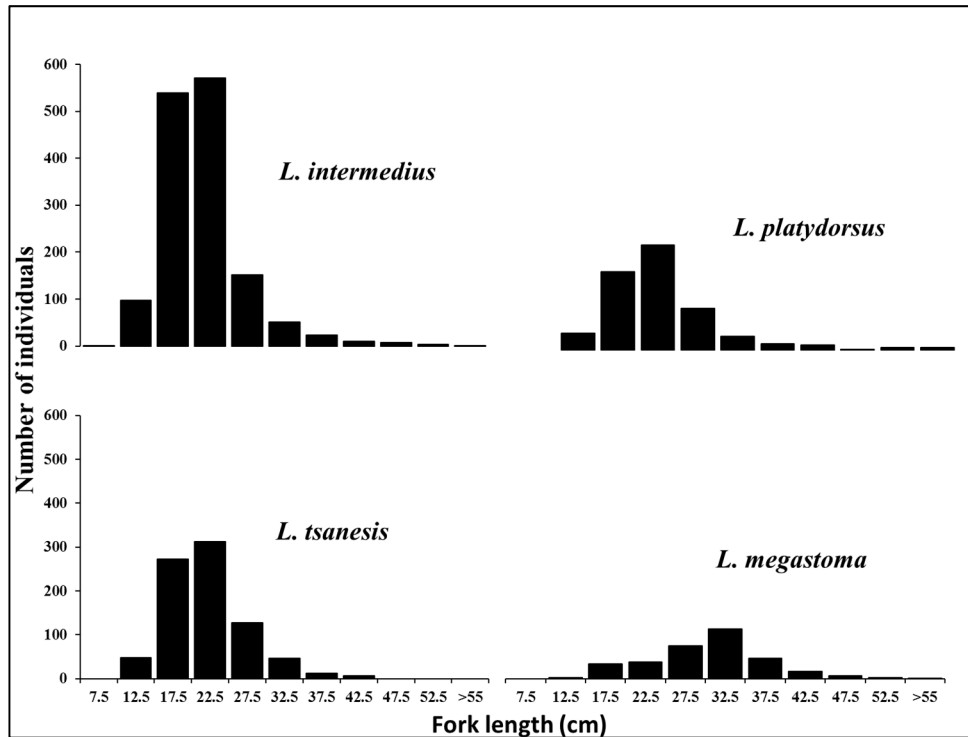

**Figure 4.** Length frequency distribution of the four dominant *Labeobarbus* spp. in Lake Tana.

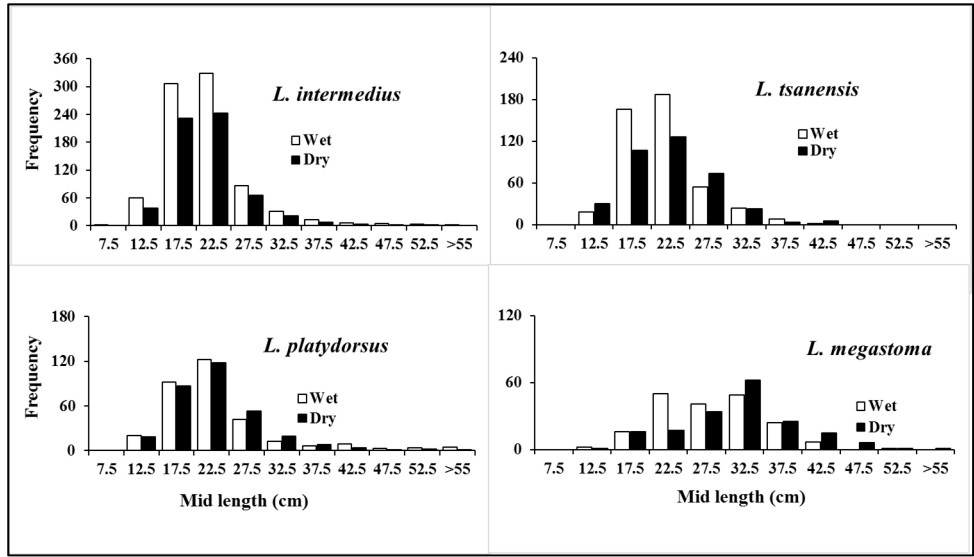

**Figure 5.** Length-frequency distribution of *L. intermedius*, *L. tsanensis*, *L. platydorsus*, and *L. megastoma* during the wet and dry seasons.

3.2.2. Size at Aaturity

The relation between proportion mature fish (specimens with ≥ IV maturity stage) and fork length was computed for the four dominant *Labeobarbus* spp. (Figure 6). Males attained their first maturity at smaller fork length than females. Size at 50% maturity ($FL_{50\%}$) for *L. megastoma* was relatively high (female = 29 cm and male = 25 cm) compared to the other three dominant *Labeobarbus* spp. (*L. intermedius*, female = 24 cm and male = 17 cm, *L. tsanensis*, female = 22 cm and male = 18 cm, *L. platydorsus*, female = 25 cm and male = 24 cm, Appendix B). Almost all of the specimens less than or equal to their size at 50% maturity were captured by gillnets with ≤ 8 cm stretched mesh size (Appendix C).

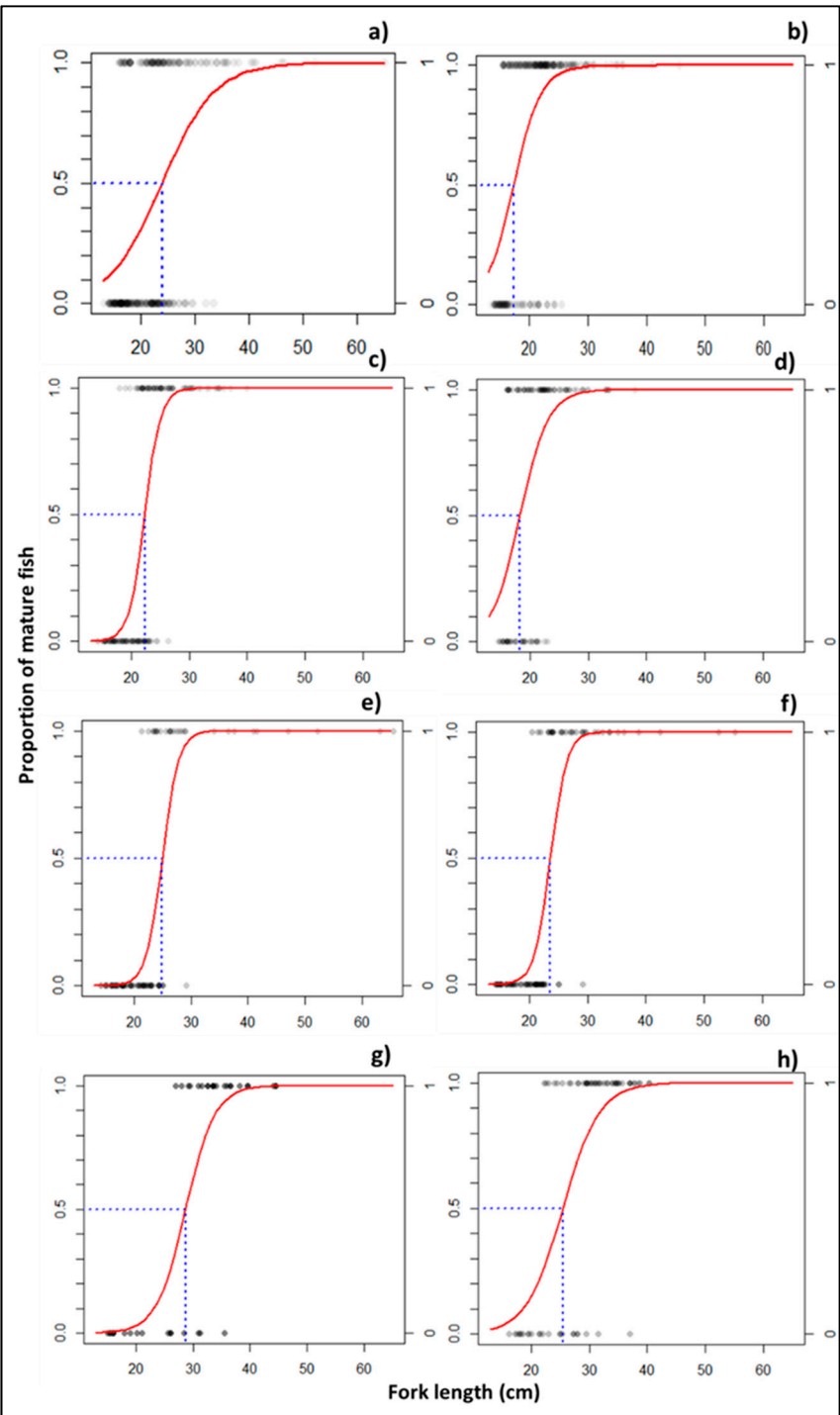

**Figure 6.** Length related to proportion mature for *L. intermedius* (**a** = female and **b** = male), *L. tsanensis* (**c** = female and **d** = male), *L. platydorsus* (**e** = female and **f** = male), and *L. megastoma* (**g** = female and **h** = male) during the breeding months (July to October) of 2016. Red lines are the fitted logistic regression; darker dots indicate that more individuals were plotted for that combination of maturity and fork length. The blue dotted line indicate the fitted size at the first maturity.

## 4. Discussion

### 4.1. The Declining Stock of Labeobarbus Spp.

Conservation of fish communities requires information for the distribution and abundance of species in a particular area [23]. The management of fish species has become a central issue of conservation biology because of environmental degradation leading globally to habitat loss and spawning ground fragmentation. This is particularly evident for endemic *Labeobarbus* spp. in Lake Tana. Due to their migratory and spawning aggregation behaviors [4] and specialized endemic characteristics [6], *Labeobarbus* spp. are susceptible to environmental degradation. Anthropogenic pressures such as illegal fishing, damming, sand mining, and agriculture are generally the main causes of the drastic decline in diversity and abundance of the *Labeobarbus* spp. [10,11,13].

In the present study, the highest catch was 130 kg/trip recorded in July and September, which is approximately equivalent to a monthly catch of 3900 kg. However, the highest catch was 24,000 kg recorded in September 1991–1993 [24] and 15,000 kg recorded in August 2001 [25], which is approximately equivalent to 2000 kg/trip and 1250 kg/trip, respectively. During the previous studies, data were collected from the commercial fisheries which use nylon multifilament gillnets. Unlike the previous studies, we used four small mesh-sized gillnets. Additionally, the average number of gillnets per trip was 19 in the previous studies, while it was 15 in this study. However, the type, length, and mesh size of the majority of gillnets were the same as those used in the previous studies. Gillnets were also set overnight as done in the previous studies, and samples were collected from similar fishing grounds. Although it is hard to make a direct comparison between the catch of *Labeobarbus* spp. here and the previous studies due to variations in sampling design, it is hard to believe that these large differences, {85% (20,400 kg) since 1991–1993 and 76% (11,400 kg) since 2001} are solely due to the small variation in sampling design. Furthermore, one can expect a high catch of *Labeobarbus* spp. in the present study since we used small-sized gillnets. Our results highlight the effect of the current illegal fishing, the rapid increase of fishers, and motorized boats, and ongoing environmental degradation on the catch of the Lake Tana fisheries. In addition, the mean CPUE for all *Labeobarbus* spp. of commercial interest has sharply declined within two decades. For example, the mean CPUE was 63 kg/trip in 1991–1993 which has markedly decreased to 28 kg/trip in 2001 [12], and 6 kg/trip in 2010 [14]. The CPUE in the present study was 2 kg/trip, which indicates that the sharp decline is still continuing. Except at the Gumara site, the CPUE of the dominant *Labeobarbus* was ≤ 4 kg/trip. The slightly higher CPUE at the Gumara site might be due to large spawning aggregations of migratory *Labeobarbus* spp. at the river mouth, as this river provides a better spawning habitat [2]. Moreover, despite the fact that the majority of *Labeobarbus* spp. are highly valued fishery resources, only five species were dominant in the current study and contributed 88% of the total catch. *Labeobarbus* spp. such as *L. acutirostris* and *L. macrophtalmus* were among the most abundant species in the 1990s and 2000s [24,25], while these species are rarely recorded in the present study. Few specimens of *L. dainellii*, *L. gorguari,* and *L. longissimus* were recorded and were entirely absent at some study sites, while *L. osseensis* was totally absent. The drastic decline in abundance and CPUE of the dominant *Labeobarbus* spp. at the majority of the sampling sites suggests that stock reduction occurred for all *Labeobarbus* and not only for those rarely recorded. The drastic catch reduction of the *Labeobarbus* species could also likely be affected by climate change.

Stock reduction of *Labeobarbus* spp. was repeatedly reported in previous studies [12,13,25]. Likewise, *Labeo altivelis*, which contributed 40–60% of the commercial catch in 1940, has declined to < 3% of the total catch within four years during its spawning migration upstream in the Luapula River [26]. This illustrates the vulnerability of the migratory cyprinid species when illegal fishing targets the spawning aggregations. Therefore, if the current situation in Lake Tana continues to deteriorate, the extinction of some *Labeobarbus* spp. appears inevitable. This will have serious implications for the lake ecosystem and the livelihoods of the local people. However, for a detailed investigation of

the spatiotemporal distributions of the *Labeobarbus* species, more research that takes into account the habitat and water depth is required.

*4.2. Insights into Size Distributions to Optimize Sustainable Fisheries Management*

Information on length distribution and size at maturity provides insights into the dynamics of the fish population [27]. Size structure, a commonly used assessment tool and that reflects the dynamic rates of recruitment, growth, and mortality. Fishing pressure has a negative effect on the size distribution and maturation [28,29]. The absence of small size classes, for example, indicates recruitment deficiency, while a lack of large size classes suggests a high mortality of adult fish [27]. To understand the status of the fisheries, understanding the length structure of fish is crucial. For sustainable fishery, fishing gears should only catch the large mature fish, while allowing immature fish ($\leq FL_{50\%}$) to escape.

However, in Lake Tana, fishing activities are illegal, unregulated, and unreported (IUU). Consequently, fishers use legally banned monofilament gillnets imported from Egypt through Sudan. Due to their small mesh size, very thin threads, and low visibility, these nets effectively harvest small-sized fishes and have adverse effects on the size composition of *Labeobarbus* spp. The $FL_{50\%}$ values we observed in the present study for the four dominant *Labeobarbus* spp. were lower than in previous findings [30]. Of the total fish caught during the breeding time of the species (July to October), 66% of *L. intermedius*, 49% of *L. tsanensis*, 38% of *L. platydorsus*, and 29% of *L. megastoma* were below the $FL_{50\%}$, and this implies that the fishes are removed before they spawn. Furthermore, during the spawning months, farmers traditionally catch *Labeobarbus* spp., on the upstream spawning grounds using a variety of destructive fishing techniques including barriers, cast nets, hooks, and poisoning (using the dried and crushed seeds of the birbira tree, *Milletia ferruginea*) [31,32]. These activities remove the spawning biomass before breeding, increase mortality of the adult fish, and result in a drastic reduction in the number of recruits. Despite large size specimens being dominant in the late 2000s [1,6], only 39 (12% of the total catch) and 26 (4% of the total catch) specimens of *L. megastoma* and *L. platydorsus* with ≥ 40 cm FL, respectively were recorded during the current study. This clearly demonstrates the expected negative effect of the destructive fishing activities in the lake and its tributaries. Moreover, in the current study, the majority of the adult *L. platydorsus* were within the length group of 20–25 cm, while they were within the length group of 27–31 cm in 1999–2003 [25].

*4.3. Management Strategies*

Fisheries management intends to find ways to protect fish resources. To overcome the danger imposed on the Ethiopian fishes, the Federal Government of Ethiopia has ratified the fisheries' proclamation in 2003. The Amhara Regional National State, where Lake Tana is located, was the first to ratify its regional fisheries proclamation in the same year. Both the federal and regional proclamations supported different management strategies, such as licensing fishermen, closed season (June to July) and areas, mesh size limitation according to biological limits (>8 cm stretched mesh), and prohibition of illegal gears, including monofilaments, beach seining, cast net, and poisonous chemicals. However, 13 years have passed without the implementation of the proclamations. On top of this, the regional government has approved the Lake Tana fisheries co-management strategy in 2013. A management committee consisting of individuals from fishers, village administrators, police, and elderly people was established. Nevertheless, similar to the proclamations, this is also not yet functional. As a result, fishing activity in Lake Tana is continuing to be illegal, uncontrolled, and unreported (IUU). Hence, the endemic *Labeobarbus* spp. are in danger of extinction, and different development pressures and resource degradations cast doubt on the sustainability of the lake's ecosystem. This can be best illustrated by the sharp reduction in abundance and negatively skewed length distribution of the *Labeobarbus* spp. Therefore, it is now time to revise and implement the proclamations.

During the 2000s, a closed season at river mouths and upstream during the peak breeding months (August and September), without restriction in other areas, was recommended [3]. During this period,

fishing at sub-littoral and offshore areas of the lake was insignificant, and the majority of the fishers used multifilament gillnets with ≥ 10 cm mesh size. Fishing targeting the spawning aggregations was the major challenge. Furthermore, during the 2000s, the number of fishers did not exceed 400, and the fishing activities were mainly limited to the shore area of the Bahir Dar Gulf and to the mouth of the Gumara River [25]. Bahir Dar was the central landing site, and almost all the catch was landed here, which enabled good communication with fishers and a clear overview of ongoing activities.

Currently, the situation has changed substantially. The number of fishers has increased more than 10-fold and their dependence on the lake fishery steeply increased. The lake fishery is currently the basis for food, employment, and income for approximately 6000 fishers. While there were not more than 10 motorized boats in the 2000s, currently, about 20% of the fulltime fishers own a motorized boat [7], and fishing activity is expanding throughout the lakesides including offshore. Above all, almost all fishers use monofilament gillnets having < 8 cm mesh size [13], and there are about 20 major landing sites across the lake, which makes it difficult to gather detailed information about the lake fisheries. In addition, the traditional fishing activities in the upstream spawning grounds and development activities in the lake catchment are intensifying. Therefore, under such complex situations, implementing a closed season (August–October) at the river mouths and upstream parts of more than 15 major tributaries will be difficult and will make fisheries management and monitoring complex. In addition, unless alternative livelihoods are created, which could be difficult for the local government, a closed season is detrimental to fishers. In sum, it is clear that the Lake Tana fisheries that target the endemic migratory *Labeobarbus* spp. will be very different in 20 years' time, and that suggested management will need to adapt accordingly.

At present, the effect of monofilament gillnets (stretched mesh size < 8 cm) negatively affects the fish populations. During the present study, nets with a mesh size ≤ 8 cm caught most of the specimens of the *Labeobarbus* spp. and their length was less than the $FL_{50\%}$. Hence, as almost all fishers in Lake Tana use mesh size ≤ 8 cm [13], it is clear that mesh size limitation is vital for the survival of Lake Tana fisheries. Mesh size restriction could also be more practical for sustainable use of the lake fish resources than the closed season. Mesh size limitation, allows fishers to undertake their activities at all times and areas and hence would not disturb their income. It is likely that in the first years of the implementation, incomes are relatively low. However, in the long term, the livelihoods of fishers would become more sustainable, since catching fish ≤ $FL_{50\%}$ would be less likely, which would allow the stock to recover. As different *Labeobarbus* spp. have different $FL_{50\%}$ [30], using a stretched mesh size of ≥ 10 cm is most advisable. In addition to mesh size restriction, we suggest the following management options for successful conservation of the lake fish and fisheries: (1) limitation of the number of fishers and fishing gears, (2) application of a hybrid bottom-up and top-down approach to identify problems, make decision and enforce the regulations [8], (3) limitation of the amount of landing sites, (4) provision of license to all fishers, (5) hire fishery inspectors, (6) establish standard data acquisition protocols and gather fisheries information regularly, (7) revision (e.g., incorporate development issues such as sand mining and damming) and proper implementation of regulations, and (8) continuous monitoring and assessment. Therefore, we suggest the local fisheries authorities incorporate these management options in the Lake Tana fish and fisheries management plan.

**Author Contributions:** S.G. conceived the main idea, collected and analyzed the data, and wrote the manuscript. A.G., S.B., W.A., and P.G. reviewed and edited the manuscript.

**Funding:** This work was supported by the Critical Ecosystem Partnership Fund (CEPF) and special research fund (BOF), Ghent University.

**Acknowledgments:** We are grateful to Birhanu Gedif, Bahir Dar University for assistance in the preparation of the study area map and Koos (J.) Vijverberg, Netherlands Institute of Ecology (NIOO-KNAW) for commenting on an earlier version of the manuscript.

**Conflicts of Interest:** The authors declare no conflict of interest. The funders had no role in the design of the study; in the collection, analyses, or interpretation of data; in the writing of the manuscript, and in the decision to publish the results.

## Appendix A

**Table A1.** Distribution of *Labeobarbus* spp. across the four study sites in Lake Tana based on Chi square test. Numbers in parentheses are expected values and different subscript letters (i.e., a, b, c) in the same row indicated significant difference in species distribution.

| Species | | Site | | | | Total |
|---|---|---|---|---|---|---|
| | | **BD** | **GU** | **GO** | **KU** | |
| *L. intermedius* | Count | 373 (405)[a] | 514 (418)[b] | 300 (343)[a] | 267 (289)[a] | 1454 |
| | Adjusted | −2.3 | 6.9 | −3.3 | −1.8 | |
| *L. tsanensis* | Count | 242 (233)[a] | 216 (241)[a] | 217 (197)[a] | 162 (166)[a] | 837 |
| | Adjusted | 0.8 | −2.1 | 1.8 | −0.4 | |
| *L. platydorsus* | Count | 119 (173)[a] | 126 (179)[a] | 237 (147)[b] | 141 (124)[c] | 623 |
| | Adjusted | −5.3 | −5.1 | 9.2 | 1.9 | |
| *L. megastoma* | Count | 100 (94)[a] | 147 (97)[b] | 69 (80)[a] | 23 (67)[c] | 339 |
| | Adjusted | 0.7 | 6.2 | −1.5 | −6.3 | |
| *L. brevicephalus* | Count | 181 (124)[a] | 63 (128)[b] | 69 (105)[b] | 133 (89)[a] | 446 |
| | Adjusted | 6.4 | −7.2 | −4.3 | 5.6 | |
| *L. crassibarbis* | Count | 32 (41)[a] | 42 (42)[a, b] | 49 (35)[b] | 24 (29)[a, b] | 147 |
| | Adjusted | −1.7 | 0.0 | 2.8 | −1.1 | |
| *L. nedgia* | Count | 43 (27)[a] | 14 (28)[b] | 15 (23)[b, c] | 26 (20)[a, c] | 98 |
| | Adjusted | 3.6 | −3.2 | −2.0 | 1.7 | |
| *L. gorgorensis* | Count | 29 (22)[a] | 27 (22)[a] | 14 (18)[a] | 8 (16)[a] | 78 |
| | Adjusted | 1.9 | 1.2 | −1.2 | −2.1 | |
| *L. truttiformis* | Count | 9 (19)[a, b] | 37 (20)[c] | 5 (16)[b] | 17 (14)[a, c] | 68 |
| | Adjusted | −2.7 | 4.7 | −3.2 | 1.1 | |
| *L. surkis* | Count | 29 (20)[a, b] | 7 (21)[c] | 10 (17)[b, c] | 26 (14)[a] | 72 |
| | Adjusted | 2.4 | −3.6 | −2.0 | 3.5 | |
| *L. longissimus* | Count | 9 (8)[a] | 9 (8)[a] | 5 (7)[a] | 5 (6)[a] | 28 |
| | Adjusted | 0.5 | 0.4 | −0.7 | −0.3 | |
| *L. macrophtalmus* | Count | 9 (7)[a] | 8 (8)[a] | 5 (6)[a] | 4 (5)a | 26 |
| | Adjusted | 0.8 | 0.2 | −0.5 | −0.6 | |
| *L. acutirostris* | Count | 0 (4)[a] | 7 (4)[a] | 2 (3)[a] | 4 (3)[a] | 13 |
| | Adjusted | −2.2 | 2.0 | −0.7 | 1,0 | |
| *L. gorguari* | Count | 3 (1)[a] | 0 (1)[a] | 0 (1)[a] | 1 (1)[a] | 4 |
| | Adjusted | 1.5 | −1.1 | −1.0 | 0.6 | |
| *L. daniellii* | Count | 1 (1)[a] | 0 (1)[a] | 1 (1)[a] | 0 (1)[a] | 2 |
| | Adjusted | 0.7 | −0.9 | 0.9 | −0.7 | |
| | Total | 1179 | 1217 | 998 | 841 | 4235 |

## Appendix B

**Table A2.** Estimates of the parameters of the length at maturity ($FL_{50\%}$) curves for both female and male of the dominant *Labeobarbus* spp. in Lake Tana in the period of May 2016 until April 2017. ** indicates significant difference at 99% confidence level.

| Species | Sex | n | a | b | *P* Value | FLmin | FL50% | 95%CI Lower | 95%CI Upper |
|---|---|---|---|---|---|---|---|---|---|
| *L. intermedius* | F | 361 | −5.0 | 0.2 | ** | 13.3 | 24.0 | 25.5 | 26.6 |
|  | M | 298 | −7.5 | 0.4 | ** | 13.5 | 17.3 | 16.5 | 17.9 |
| *L. tsanensis* | F | 183 | −16.5 | 0.7 | ** | 14.2 | 22.3 | 21.9 | 22.9 |
|  | M | 125 | −7.6 | 0.4 | ** | 14.5 | 18.3 | 17.2 | 19.3 |
| *L. platydorsus* | F | 108 | −17.7 | 0.7 | ** | 14.1 | 25.0 | 23.8 | 25.9 |
|  | M | 93 | −17.2 | 0.7 | ** | 14.2 | 23.6 | 22.6 | 25.2 |
| *L. megastoma* | F | 39 | −11.4 | 0.4 | ** | 15.0 | 28.7 | 26.2 | 31.0 |
|  | M | 70 | −8.1 | 0.3 | ** | 16.2 | 25.4 | 23.2 | 27.3 |

## Appendix C

**Table A3.** Catch composition of the *Labeobarbus* spp. in Lake Tana using different mesh size of gillnets. FL is fork length (cm), TW is total weight (g) and N is number of individuals.

| Species | Mesh Size (Monofilament Gillnets) | | | | | | Mesh Size (Multifilament Gillnets) | | | | | | | | | | | | | | |
| | 4 cm | | | 6 cm | | | 6 cm | | | 8 cm | | | 10 cm | | | 12 cm | | | 14 cm | | |
| | FL | TW | N | FL | TW | N | FL | TW | N | FL | TW | N | FL | TW | N | FL | TW | N | FL | TW | N |
|---|---|---|---|---|---|---|---|---|---|---|---|---|---|---|---|---|---|---|---|---|---|
| *L. intermedius* | 10–25 | 11–413 | 438 | 13–42 | 30–1060 | 272 | 14–34 | 13–434.9 | 209 | 14.7–36 | 30–1255 | 357 | 18–52 | 80–2800 | 72 | 18.3–46 | 60–1625 | 82 | 22–65 | 150–4305 | 24 |
| *L. tsanensis* | 13–30 | 33–419 | 276 | 12–30 | 32–425 | 142 | 13–34 | 32–489 | 121 | 16–38 | 24–845 | 195 | 18–37 | 80–2755 | 60 | 20–43 | 82–1150 | 42 | 43 | 1325 | 1 |
| *L. platydorsus* | 13–26 | 31–232 | 150 | 12–28 | 40–315 | 99 | 12–29 | 17–305 | 88 | 16–47 | 45–1830 | 174 | 16–59 | 45–2685 | 68 | 21–52 | 147–2125 | 23 | 26–66 | 470–4095 | 21 |
| *L. brevicephalus* | 10–23 | 10–165 | 280 | 13–23 | 30–160 | 63 | 13–25 | 34–185 | 49 | 14–26 | 35–1026 | 53 | 28 | 340 | 1 | | | | | | |
| *L. megastoma* | 15–22 | 37–106 | 22 | 18–25 | 65–150 | 7 | 16–33 | 35–420 | 49 | 12–35 | 114–448 | 70 | 22–45 | 102–1070 | 112 | 27–57 | 170–2300 | 48 | 34–55 | 405–2175 | 31 |
| *L. crassibarbis* | 14–22 | 39–145 | 34 | 13–27 | 35–260 | 19 | 13–26 | 35–256 | 15 | 17–60 | 65–3660 | 42 | 27–55 | 215–2550 | 11 | 19–44 | 72–1485 | 18 | 37–53 | 1200–2330 | 8 |
| *L. gorgorensis* | 14–23 | 43–180 | 27 | 20–27 | 100–275 | 12 | 21–25 | 100–225 | 9 | 21–33 | 123–420 | 8 | 21–31 | 135–460 | 10 | 24–43 | 179–1300 | 8 | 33–40 | 556–1060 | 2 |
| *L. longissimus* | 15–16 | 38–49 | 3 | | | | 21–29 | 145–390 | 5 | 21–33 | 180–625 | 8 | 21–45 | 115–1065 | 7 | 35–41 | 620–940 | 3 | 32–43 | 386–1010 | 2 |
| *L. nedgia* | 13–29 | 30–355 | 46 | | | | 12–29 | 22–316 | 18 | 18–27 | 57–289 | 13 | 26–41 | 200–1060 | 9 | 21–55 | 133–3060 | 7 | 25–56 | 218–3190 | 4 |
| *L. macrophtalmus* | 15 | 51 | 1 | | | | 19–20 | 95 | 2 | 21–28 | 120–279 | 8 | 23–36 | 158–645 | 10 | 31–37 | 266–525 | 5 | | | |
| *L. truttiformis* | 16–17 | 46–57 | 7 | | | | 21–25 | 120–580 | 9 | 23–31 | 219–445 | 13 | 27–40 | 245–1000 | 21 | 25–40 | 288–820 | 10 | 26–49 | 700–1820 | 8 |
| *L. surkis* | 14–26 | 35–413 | 51 | 17–21 | 74–125 | 4 | 17–25 | 80–238 | 13 | 18–22 | 90–161 | 4 | | | | | | | | | |
| *L. gorguari* | 18–21 | 78–121 | 2 | | | | 20 | 122 | 1 | 40 | 980 | 1 | | | | | | | | | |
| *L. daniellii* | | | | | | | 24 | 155 | 1 | 31 | 345 | 1 | | | | | | | | | |
| *L. acutirostris* | | | | | | | 21–24 | 95–150 | 2 | 26–28 | 215–285 | 5 | 26–31 | 270–340 | 3 | 37–39 | 710–780 | 2 | 54 | 2320 | 1 |

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
