# Peer review of "The Endemic Species Flock of Labeobarbus spp. in L. Tana (Ethiopia) Threatened by Extinction: Implications for Conservation Management"

_water, doi:10.3390/w11122560_

Round 1

Reviewer 1 Report

This is a well written article on a topic that should be of high interest to readers because of the decline in catch of several endemic cyprinids in an Ethiopian Lake. The authors present sufficient detail to document a severe decline in catch for multiple species. Less clear is the cause of the decline which is presented as both illegal fishing and environmental degradation. The increase in the number of fishermen and their increased use of motor boats does seem correlated. Perhaps climate change is also causing this decline and should at least be mentioned as a possibility. 

I think the writing is a bit wordy and edited the manuscript to suggest changes to reduce wordiness. I have attached an edited version for the authors to consider. I think the article is acceptable with these revisions.

Author Response

 Thank you very much for the detail corrections, which are valuable to improve the quality of the manuscript. We have thoroughly incorporated them and indicated by track change in the main text.

Reviewer 2 Report

This is in principle an interesting study, based on good sample sizes. The results - when contrasted to historical records - paint a fairly depressing picture what is going with the fishery of the species in the focus of this study. Hence, while I believe these results should be published, there is room for improvement before this is done. Some suggestions in the order they appear in the ms follow.

P3. It is not clear from here that the main gear used were not monofilament. Please explain what kind of gear was used to catch the larger fish. Why did you use different kind of gear to catch smaller sized fish?

P4L119. What kind of Bonferroni method? Sequential? Please specify.

P4.L120. This is a ratio of ratios. Ratios themselves are often problematic, but If you have ratio of ratios, I am not sure what kind of distribution they are expected to follow. Can you please convince the readers that this metric is meaningful.

L138. Please define Stage IV.

L 141. This equation can expressed in the main text. More complex equations have been done so - why a serate line(s) for this?

L.144. I did not understand "including the family=binomial" argument". Please explain.

L. 164.df =42. I do not understand how it is possible that you have so many degrees of freedom? Did you consider all sampling occasions independent? If so, I do not thinks this is justified. You should control for this in statical analyses.

L178-179. All abbreviations in the table headings should be explained.

L182. All the the statical interpretation in this paragraph should ne revised. Only P-values are given now. At minimum, test-statistics and degrees of freedom should be given too.

L203. We cannot see statistics for this.

L291-292. But one could argue also the opposite: taking out the largest fish could be the most detrimental action because they are the most fecund, and important for production of young?

L294. If I have not overlooked something, this is first time we learn monofilaments are banned. 

L340. Can you back up this with a reference?

Author Response

Thank you very much for your remarks. We incorporated all comments accordingly. We have also provided a response to each question and kindly attached the document herewith. 

Round 2

Reviewer 2 Report

The authors have responded to my earlier comments appropriately and done the suggested changes. No further comments.